# The Synergistic Process of Improvement in Cognitive Behavioral Therapy for Major Depression

**DOI:** 10.3390/ijerph18052292

**Published:** 2021-02-26

**Authors:** Anders Malkomsen, Jan Ivar Røssberg, Toril Dammen, Theresa Wilberg, André Løvgren, Julie Horgen Evensen

**Affiliations:** 1Division of Mental Health and Addiction, Oslo University Hospital, 0424 Oslo, Norway; j.i.rossberg@medisin.uio.no (J.I.R.); UXTHWI@ous-hf.no (T.W.); andre.lovgren@medisin.uio.no (A.L.); 2Institute of Clinical Medicine, University of Oslo, 0318 Oslo, Norway; 3Department of Behavioral Science in Medicine, Faculty of Medicine, University of Oslo, 0372 Oslo, Norway; Toril.dammen@medisin.uio.no; 4Nydalen Outpatient Clinic, 0424 Oslo, Norway; uxevej@ous-hf.no

**Keywords:** improvement, depression, cognitive behavioral therapy, patients’ perspective, qualitative study

## Abstract

Background: There is a substantial lack of qualitative research concerning individual cognitive behavioral therapy (CBT) for patients with major depressive disorder (MDD). In the present study, we wanted to explore how patients suffering from MDD experience improvement in CBT. Method: Patients with MDD (*N* = 10) were interviewed at therapy termination with semi-structured qualitative interviews. The transcripts were analyzed using a thematic analysis approach. Results: We identified three elements that were relevant to the process of improvement for all patients: the therapeutic relationship, the therapeutic interventions and increased insight. There is a dynamic interrelationship and synergy between these elements that may explain why patients considered the same elements as helpful, but often in different ways and at different stages of therapy. Conclusions: Highlighting the synergies and interrelationship between the elements that patients experience as helpful, may help therapists to learn from and utilize these experiences. This is a reminder of the importance of always being attentive to the individual processes of patients.

## 1. Introduction

Major depressive disorder (MDD) is expected to rank first in overall disease burden in high income countries by the year 2030, measured in disability-adjusted life years [1]. Effective methods to prevent episodes of major depression, or to improve the volume or quality of treatment, are highly needed to reduce the burden of the disease [2]. Meta-analyses of psychological therapies for depression show that several forms of psychotherapy can reduce symptoms of depression [3,4,5]. Cognitive behavioral therapy (CBT) is the most researched psychotherapy for treating depressed adults and has consistently been shown at least as effective as other psychotherapies [6]. Still, some patients do not respond sufficiently to CBT and relapse is common, showcasing the need for more research on the process of improvement in CBT for major depression [6].

Qualitative research addressing patients’ experience of psychotherapy has grown in the last decades, resulting in two qualitative meta-analyses and a qualitative meta-synthesis [3,4,5]. A study by Levitt et al., which also included several other approaches to therapy besides CBT, shows the many factors in therapy that contribute to improvement [4]. They built upon the foundation laid by the meta-analysis by Timulak et al. [3] and grouped their findings into five clusters with numerous subcategories. Since so few of the studies focused solely on one therapy orientation, all change mechanisms from different orientations are intertwined. The first cluster emphasized the process of change through stimulating curiosity, engagement in pattern identification and narrative reconstruction. The second cluster emphasized how a caring, understanding and accepting therapist allows patients to internalize positive messages and develop self-awareness. The third cluster emphasized how a professional structure creates credibility, safety and clarity, but also casts suspicion on the authenticity of the therapeutic relationship. The fourth cluster emphasized how therapy develops as a collaborative effort with discussion of demographic differences between the patient and therapist. The final cluster emphasized how recognition of the client’s agency allowed for more engagement and interventions that fit the client’s needs.

A qualitative metasynthesis by McPherson et al. of different psychological therapies for depression, including CBT, identified several elements that contributed to improvement [5]. A common key process of improvement referred to by many patients was that of gaining new insights to help reframe emotions or problems, and to identify existing coping mechanisms. In cognitive approaches they found that patients often improved by learning techniques for managing feelings, which later became naturalized and automatic, but they do not specify which techniques were used. Across all approaches it appeared that different people find different elements helpful, hence they conclude that therapy needs to be more individualized.

Timulak has found that views on what is significant in therapy often differs, with the therapist’s and patient’s perspectives only matching in approximately 30–40% of cases [7]. In general, patients value the relational and emotional aspects of therapy more than the therapists, who tend to favor the cognitive aspects. Timulak also found that a match between the patients’ and the therapists’ perspectives on therapy increases with a good outcome and relationship. If therapists are more attuned to the patients’ experiences in therapy, the therapy may be more successful [7]. Thus, understanding how patients experience improvement from depression in CBT has the potential to advance the theoretical understanding of the change processes and improve outcomes [8]. Although the quantity of qualitative research is growing, there is still a substantial lack of qualitative research concerning face-to-face individual CBT for depressed adult patients.

Nilsson et al. interviewed 32 patients, four of whom were men, who received either CBT or psychodynamic therapy (PDT) about how they experienced change during therapy [9]. The patients’ formal diagnoses were not known, but they were recruited from a clinic treating a range of both neurotic and psychotic disorders. The researchers found several common experiences in the group of satisfied patients, like learning new coping tools, especially understanding and coping with feelings in a way that reduced their levels of anxiety. Patients in both therapy groups emphasized the importance of a personal chemistry, feeling that they could rely on the therapist and that the therapist was able to create structure in the sessions. It was notable that the satisfied CBT patients reported more often than the PDT patients that their problems had been normalized and that therapy made them more able to cope with difficult situations. Facing fears and exposure techniques were experienced as very effective by patients receiving CBT, and normalization seemed to be an important mechanism. Other specific therapeutic interventions that were found to be useful in CBT were homework assignments and mastery of relaxation techniques.

De Smet et al. interviewed 48 improved or recovered patients with depression who received time-limited CBT and PDT to find out what “good outcome” meant in the patients’ own experiences [10]. This study, however, analyzed data from the CBT and PDT groups together, and did not investigate experiences specific to CBT. They found that patients in both groups felt they improved when they learned to care for themselves and increased their self-understanding, and that patients in the CBT-group more often reported that they learned new coping skills. The patients who felt they had recovered often described a feeling of empowerment, insight, personal balance and better interpersonal relationships. They found big differences between the reduction of depressive symptoms as assessed by a standard self-report questionnaire (Beck’s depression inventory (BDI)) compared to patients’ actual experiences of improvement, and conclude that improvement in depression require a multidimensional understanding.

De Smet et al. also studied the processes underlying “good outcome” in the same group of patients [11]. They grouped their analysis of the therapeutic process into four categories, but did not differentiate between PDT and CBT in their categorization. Firstly, they found that therapy stimulates a different perspective via the general process of talking, therapist questioning and reflection. Secondly, patients experienced that being actively involved in therapy promoted healing. Thirdly, the patients felt empowered by the therapists who acknowledged their suffering and reinforced their growth, thereby creating a safe haven that seemed to set the right condition for engaging in therapy. Lastly, the patients also contributed their healing to influences outside of the therapy room, like being supported by significant others and the positive response they received from others on gains and changes achieved in therapy. However, they observed large differences within the groups, which challenge the idea that all patients move towards a similar outcome in a linear trajectory of change.

As shown, some qualitative studies already exist on the topic of what works and does not work in CBT. However, the literature is still relatively scarce. A thorough evidence-based explanation of how therapeutic processes produce change is lacking [12,13]. Process outcome studies into the mechanisms of change in CBT have found cognitive change and compensatory skills to be contributing to improvement, but how related therapeutic techniques are experienced by patients, and how the different techniques and elements of therapy influence each other, remains unexplored [14]. Our study aims to reduce this gap in our body of research by exploring how ten adult patients suffering from major depression experienced improvement in manualized time limited CBT, using sampling control, manualized treatment, supervision of the therapists and treatment fidelity control. Our research question was: What is experienced by patients as the important elements in the process of improvement, and what do the patients say about the elements in this process?

## 2. Materials and Methods

### 2.1. Design, Ethics and Data Collection

The present study took place at two public psychiatric outpatient clinics in Oslo, where patients with a wide range of mental illnesses are treated. These clinics are part of the secondary care system and require that patients are referred by a doctor, often a general practitioner. The study is part of the ongoing Norwegian project on the Mechanism of Change in Psychotherapy (MOP) [15]. The overall aim of MOP is to examine moderators and mediators in CBT and PDT and develop knowledge about what works for whom and how. The participants are randomized to either CBT or PDT; the CBT consisting of 16 weekly sessions and three monthly booster sessions (total of 19 sessions). Clinical assessments are conducted at baseline, during therapy, at the end of therapy and at follow-up investigations 1 and 3 years after treatment termination. Inclusion and treatment in the MOP-project is still ongoing, so the outcome data for the participants are not yet available. The Central Norway Regional Ethics Health Committee (REC South East 2016/340) approved the MOP-study, including the qualitative interviews. Clinical Trial gov. Identifier: NCT03022071. Written informed consent was obtained from all participants.

### 2.2. The Interviews

All patients from the initial phase of the study were invited to a qualitative in-depth interview after completing therapy. A few weeks after the end of therapy the second author conducted these interviews with a focus on patients’ experiences with treatment, the therapeutic processes and therapeutic gains. The interviews had a duration of 45–60 min and took place at the outpatient clinic where the patients had received therapy. We aimed for an informal and supportive tone, using semistructured interviews and encouraging the participants to elaborate on themes of relevance. The patients were questioned about therapy in general, what had been helpful and not helpful, how therapy affected their relationships and to what extent they could use anything from therapy in their everyday life. The structured questions are found in Table 1**.** A research assistant transcribed the interviews and anonymized all the transcriptions.

### 2.3. Participants

Ten participants were included in this qualitative study. Inclusion criteria were fulfilling the criteria of MDD according to the Diagnostic and Statistical Manual of Mental Disorders, 4th Edition (DSM-IV) (based on a clinical interview and The Mini-International Neuropsychiatric Interview (MINI)), age 18–65 years, the ability to understand, write and speak a Scandinavian language, and willingness and ability to give informed consent. Exclusion criteria were current or past neurological illness, traumatic brain injury, current alcohol and/or substance dependency disorders, psychotic disorders, bipolar disorder type 1, developmental disorders and mental retardation. Mean age at inclusion of the interviewed persons, six females and four males, were 36.5 years (range 22–48). All met the criteria for MDD according to the DSM assessed by MINI [16]. Additionally, one patient was diagnosed with comorbid panic disorder with agoraphobia, post-traumatic stress disorder (PTSD) and generalized anxiety disorder, one with panic disorder with agoraphobia and generalized anxiety disorder and one with generalized anxiety disorder. Baseline level of depression was a mean of 15.4 (range 8–23) measured with the Hamilton depression rating scale, indicating the level of depression (15.4 indicating mild–moderate depression) [17]. According to the Structured Clinical Interview for DSM-IV Axis II, one patient was diagnosed with an unspecified personality disorder and one patient with avoidant personality disorder [18]. Of the ten interviewed participants, one dropped out of therapy due to dissatisfaction after seven sessions, but was still included in the study. Of the remaining nine participants, all completed therapy.

### 2.4. Therapists and Treatment

All patients had to pay for their sessions as they would normally do in the outpatient clinic. Patients paid for sessions until they reached the yearly limit of 230 euros, after which all health care is free in the Norwegian health care system. The therapists, with the exception of one psychiatric nurse, were psychiatrists and psychologists. All therapists had a minimum of two years of training in CBT. In addition, they received one year of training on the principles of CBT in the MOP study before receiving patients for therapy. The principles were based on the book Cognitive Therapy of Depression by Aaron Beck et al. [19].

Our protocol gave therapists some guidance on how to structure their therapy, as laid out in Table 2. The first sessions should focus on building a therapeutic alliance and engaging the patient. All patients should get a case formulation and write a problem list. Psychoeducation should be provided so that patients understand the principles of CBT (socialization to the model). The interventions should focus on both short- and long-term goals. Different cognitive techniques, described by Beck, should be applied to reach these goals. Homework should be given in each session. The last part of therapy should focus on consolidation and relapse prevention.

CBT aims to treat depression by changing the structuring of the depressed patients’ worldview. This can be achieved by changing their cognition, attitudes and assumptions. The underlying theoretical rationale is that an individual’s affect and behavior are determined mostly by the way he or she cognitively structures the world [19]. The basic structure consists of the patient’s thoughts about the self, the world and the future. Restructuring can happen through several different techniques. In theory, the patient learns these techniques from the therapist and then incorporates them into his everyday life.

The core healing techniques presented by Beck et al. are the monitoring of negative, automatic thoughts; examining evidence for and against these thoughts; substituting distorted thoughts with more reality-oriented interpretations; recognizing the connections between cognition, affect and behavior and identifying and altering dysfunctional beliefs [19]. Therapists in CBT are supposed to be continuously active and interacting with the patient. Therapy is structured in a way that engages participation and collaboration. The therapy is focused on the “here-and-now” and limited attention is paid recollecting the past [19].

Experienced CBT-clinicians monitored adherence to the treatment principles in weekly group supervisions throughout the therapy period. Treatment fidelity strengthens the probability that the therapists actually deliver the intended therapy. We therefore videotaped all therapy sessions, and a group of experienced CBT-clinicians made sure that all therapists adhered to treatment protocol. Video recordings from the therapy sessions were reviewed by the group with focus on the initial phase of treatment, case formulation, individual treatment strategy and termination of therapy. Few other qualitative studies of this kind run similarly strict fidelity control.

### 2.5. Analysis

The object of our study was how patients experienced the process of improvement. As patients’ experiences and opinions may diverge, we chose to integrate diverse and contrasting experiences in the results, in order to give a more nuanced picture of how patients perceived the therapy process. We used the thematic analysis approach to study the patients’ experiences of improvement in therapy [20]. Thematic analysis is a method suited to analyze the experiences and meaning-making of participants. Themes or patterns within our data were identified using an inductive “bottom up” approach, without trying to fit the data into a pre-existing frame. Our analysis progressed from a description where the data were organized to reflect the patterns and semantic content, to an interpretation where we theorized the significance, meaning and implications of the themes and patterns [21].

It is always important that the theoretical position of the analysis is made transparent [22]. We chose the hermeneutical–phenomenological position to study the patients’ experiences of improvement in therapy [23,24]. Phenomenology is the study of lived experience with emphasis on individual meaning-making [25]. The hermeneutical approach acknowledges that interpretation is both a necessary and inevitable part of trying to understand the lived experience of others, and that interpretation always happens within a concrete cultural and historical context [22].

The first author read the interviews looking for answers to the question: What is experienced by patients as the important elements in the process of improvement, and what do the patients say about the elements in this process? After several readings the first author wrote down his initial impressions and discussed them with the second author. This resulted in a reduction of the transcribed material into different categories that represented different aspects of the patients’ experiences. The second and the last author then read all the transcripts and gave feedback on the first author’s categorization. Thereafter, the second author initiated a process in which all authors met and discussed their unique understanding of the material, and criticized the first author’s categorization. The first author then reorganized the material after getting feedback from the other authors. Finally, all authors agreed on the current categorization and presentation. This optimized the validity and made our interpretations less dependent on individual preferences [22]. The authors have different therapeutic orientations: J.E., T.D. and J.I.R. are CBT therapists and T.W. is a PDT clinician. A.L. and A.M. have no specific therapeutic orientation.

## 3. Results

Our analysis resulted in the identification of three important elements in the process of improvement: (1) the relationship between therapist and patient; (2) the therapeutic interventions and (3) gaining insight. The main elements and their important aspects are summarized in Table 3. In the following text, we will describe how the patients experienced all three elements. We additionally detail how these elements work in a synergistic process. The quotes given are meant to represent both the similarities and differences between the patients. As suggested by Hill et al. we indicate the recurrence and representativeness of patients’ experiences by using the labels general, typical and variant [26]. The three main elements were mentioned by all or all but one patient and is therefore labeled as general, in the text referred to as all patients. An element is considered typical when it is mentioned by more than half the cases, in the text referred to as most patients. We used the term some patients when the element was found to be a variant represented by less than half but more than two cases. Patients were given individual numbers (#1–10) for identification.

### 3.1. The Relationship between Patient and Therapist

All the patients emphasized the importance of a good relationship with their therapist. At the same time, they differed in their views on how good the relationship had to be, and what a “good” therapeutic relationship really is. When asked what helped him the most, one patient (#1) said: “I think trust is really important, if I had to choose one thing”. Trust is described as being an essential catalyst for the process of opening up in therapy and being able to “talk about the problems you actually have”.

Some patients emphasized the importance of “chemistry”, but few could explain what this expression really meant for them. One patient (#2) explained it like this: “We weren’t similar, but we were compatible. None of us are too formal or solemn. We agreed that some things were like “you just don’t behave like that!””. This similarity made the patient feel safe, and she felt sure that the therapist would agree with her in relational conflicts because they had “the same world view”. Another patient (#3) said that chemistry with the therapist made her feel more “comfortable”. She therefore dared to “say out loud” what she was thinking.

The therapists with whom the patients experienced a good relationship were generally described as “interested”, and no other description was used by more than one patient. One patient (#4) felt that meeting an interested therapist is “more motivating, and gives me more hope of becoming well”. Another patient (#1) described how being asked questions made him feel that the therapist was curious and interested, and that the in-depth questioning made him feel understood: “I felt that he understood me … that he understood and cared (…) because he asked and dug around when he was uncertain”. One patient (#5) also mentioned the importance of an interested, but not “too emphatic” therapist, and valued that her therapist did not pity her: “I think it was good that she was helpful and calm, openly listening, without being like “poor you”. I can’t stand that. I don’t go to therapy to get pity”. Some patients also mentioned that they appreciated when the therapist managed to be both professional and personal. One patient (#2) said she felt understood by the therapist and that their relationship felt both personal and professional. She remembered vividly that the therapist had asked her about a book recommendation for her sons, and that this “made me feel not just like a patient, but as a person with a personality and abilities in addition to a mental illness”. Another patient (4#) was treated by the same therapist, but felt that the therapist was too personal, and said that “It became too friendly. It was like I was sitting at a café speaking to a friend”.

One patient (#6) said the therapist reminded him about his mother, and that “I kind of expected to get the same answers from her that I got from my parents”. Then the therapist responded quite different from his mother in a particular situation: “My mom and dad were like “now you must finish therapy and try to be single for a while before you meet someone new”. (…) But then the therapist said: “Yes, but why can’t you do both?” And I did. Then I felt, it was like, everything kind of loosened a bit”. The patient says this was the moment of therapy he most vividly remembers because he was “100% sure she (the therapist) was going to say the same as my mom and dad” and “then she said the opposite”.

Some patients felt that their private relationships were strengthened by the therapy. They emphasized the importance of using their relationship with the therapist to learn and test new communication skills, like expressing sadness and vulnerability. One patient (#7), who learned how to accept and express his vulnerability, said that his ability to be more open had “saved” his relationship to his girlfriend.

Not all patients cared if they were liked by their therapist or not. One patient (#8) thought the question he got in the Working Alliance Inventory (WAI) “Do you think that the therapist likes you?” was “a very strange question because she has to like me, it’s her job” and said “it’s not about liking or not, it’s more about having chemistry”. Another patient (#9), who did not know if the therapist liked her or not, said that “I guess it doesn’t really matter for the treatment”.

### 3.2. Therapeutic Interventions

We grouped cognitive techniques, therapeutic assignments and medication together by using the term “therapeutic interventions”. The goal of a therapeutic intervention is to initiate improvement in how the client feels, thinks and behaves. We identified seven therapeutic interventions that patients mentioned as significant in their process of improvement: Socratic questioning, using the cognitive diamond and the ABC model (activating events, beliefs and consequences), discussing thinking traps, advice on behavior, homework assignments, which they were given responsibility, and medication. Some of these interventions, like Socratic questioning and the cognitive diamond, are exploratory in nature. Other interventions, like homework, advice and thinking traps, are more instructive or psychoeducative.

Most patients mentioned the way their therapist asked them questions as important. One patient (#2) quite precisely summarized the technique of Socratic questioning (without knowing it): “She kind of led me to realize things for myself, I think. More like: “Yes, but is that really true? How can you know for certain?” And then you see the strings holding the dolls inside your head”. Another patient (#3) said: “I actually feel that was the whole treatment. That I had a preconceived opinion of “this is how it is, this is how that person is”, and then the therapist asks “how can you know for sure?” or “have you asked this person about this?”. Then I become uncertain because I understand that I have just constructed a truth inside my head”. Still, some patients felt that this style of questioning was not always right for them. One patient (#8) said: “I understand that it is supposed to be a conversation where I should come to my own recognitions. But when you’re in the middle of a crisis, that’s not what you need”.

Most patients also mentioned using the cognitive diamond or ABC model as a helpful way to structure, increase awareness of and understand the relationship between thoughts, feelings, actions and bodily sensations. One patient (#1) said: “That diamond or the ABC model, it’s more like an aid or a tool (…) you get something concrete on a paper. That makes it easier to understand, quite frankly”. Another patient (#10) said: “She (the therapist) asked many questions that I needed to think about a lot to answer. Some questions I couldn’t answer, but then she could say something, or draw up a model, and it was like “Yes, that’s just the way it is””. One patient (#3) said about the cognitive diamond that “it has really stuck with me”, another (#1) said he found it helpful, but not absolutely necessary.

Some patients mentioned that it helped them improve when their therapist pointed out that they were thinking in a nonconstructive way. One patient (#2) put it like this: “She (the therapist) said that “just because you feel it doesn’t mean it’s correct””. This patient, who in addition to emotional reasoning was prone to black-and-white thinking, explained her improvement like this: “Instead of thinking in absolutes, like “I’m not an Instagram model, therefore I am ugly and fat and horrible” I instead think that “it is possible to be somewhere in between those two””. Another patient (#5) who said that her therapist helped her organize her thoughts, put it like this: “Before I tended to be very black-and-white. I jumped to conclusions, went straight to the basement. Now she (the therapist) has made the (…) frustration-phase much longer”.

Some patients stressed the importance of getting behavioral advice in therapy. One patient (#2) appreciated getting new advice each week: “I think it helped with all those advices I got, a concentrated dosage each week, like “Try this? Oh, it didn’t work? Then try this instead””. One patient (#6) even wanted more explicit demands from the therapist: “I wish the therapist had pushed me more, like “this week you have to exercise two times, or else …” Because I can’t do things on my own, I need someone pushing me”. However, not all patients expressed a need for advice, and one patient (#5) was provoked by getting behavioral advice from the therapist: “I don’t always need advice. Sometimes I just wish for someone to listen and accept my emotions in everything that’s going on”.

Most patients mentioned the homework assignment as an important part of their therapy. One patient (#1) said: “I think it was important because it forces you to do something yourself. You can’t just lean back and hope someone else fixes it”. Some patients also said that the homework made therapy more effective. One patient (#8) put it like this: “Therapy doesn’t help if you just do some work one hour per week. You have to work all the time, because it takes time to change your mentality and your automatic responses”. One patient (#4) also used her homework to cope with her anxiety: “I started using the homework as a kind of calming technique after an anxiety attack. I calmed down while filling out the forms”. There were also mixed views on the importance of homework. The main criticism from patients who did not appreciate the homework was that “it just felt like too much pressure to perform”. One patient (#10) admitted: “I think they (the assignments) would have been more important if I had spent more time working with them”. The main reasons the patients did not do their homework was that they either forgot or did not find the time to do it.

Being given autonomy and thereby responsibility for their own healing process seemed to be valued by most patients. One patient (#2) said “I liked that autonomy. It was like I was doing the work”. At the same time, some patients experienced this responsibility more like a pressure (#4): “It was too much pressure coming here because I always had to bring something to talk about. It almost felt like a job I had to prepare for”.

Some patients experienced medication as important for their ability to get the most out of therapy sessions. One patient (#5) felt she was not available for therapy before starting antidepressants (SSRI): “It was like the medication made the world less black-and-white. Then I could really start taking advantage of the forms (ABC model to analyze a situation)”. Another patient (#10) felt that the medication (SSRI) made her thoughts more “constructive”. One patient (#2) who started using non-addictive sleeping pills (melatonin) described the effect on her sleep “like magic” and felt that stabilizing sleep was essential for the effectiveness of CBT. Even though some patients mentioned medication as an important part of their healing process, there was no consensus. One patient (#8) found the effect (from SSRI) “not really good enough” and another (#7) experienced “a pretty good boost” (from SSRI), but found the CBT to be the most important part of his healing process.

### 3.3. Insight

The patients gained insight from therapy in several different ways. We identified three main ways the patients experienced an increase of insight: organizing and raising awareness of their own feelings, thoughts, bodily sensations and behaviors; gaining ability to distance themselves from and question their own thoughts and expanding their perspective by focusing more on the positive aspects of themselves.

Most patients said that therapy raised their awareness of their thoughts and feelings, and the ability to differentiate between them. One patient (#3) said: “If I suddenly feel jealous or something painful pops up inside me, I try to think “okay, but what do I think about this feeling that makes me feel so sad?””. All patients who mentioned this were asked how they got to this state of raised awareness, but all had great difficulty answering. One patient (#7) said: “I can’t say anything else than “it’s just a process””.

Some patients changed their way of thinking about their thoughts and said they experienced this change as an improvement. One patient (#3) said: “She (the therapist) helped me understand that “Ok, here comes these thoughts and there’s nothing I can do about it, but I can change the way I think about it””. Another patient (#2) internalized the questioning voice of the therapist: “I had the voice of my therapist in the back of my head saying “yes, that’s what you’re thinking now, but that doesn’t mean it’s the truth”. Then I put a little question mark behind all these... like “you’re a terrible person”–question mark”. Yet another patient (#5), bothered by constant suicidal rumination, learned to accept her thoughts instead of fighting them: “I was very ashamed of these thoughts and didn’t understand where they came from. But then she (the therapist) rendered them harmless in a way because she said it’s okay to have such thoughts”.

Some patients also said that therapy helped them recognize automatic thoughts. One patient (#6) put it like this: “The therapist said you have to stop and ask yourself “Do you really have to?” instead of just automatically saying “yes”. So, I have changed my automatic response a little bit. Now I’m much better at saying “no” and not take responsibility for things I shouldn’t”.

Some patients seemed to change their core beliefs about themselves and experienced this as an improvement. One patient (#3) put it like this: “Instead of focusing on everything bad about myself, I started becoming more aware of my positive qualities”. The patient explained that her therapist helped her by forcing her to write down positive things about herself. Another patient (#10) felt like a bad mother because she sometimes could not get her children to brush their teeth at night. She said: “They just didn’t listen. Then she (the therapist) asked “Do you read for them at night? Do you feed them? Do you help them with their homework? Do you tell them you love them?” And I answered yes to all those questions. (…) Then I started thinking about all the things I actually managed”.

### 3.4. Synergy in Therapy: Case Example

There often seemed to be an interaction between the therapeutic relationship, interventions and insight that made the therapeutic effect greater than the sum of the combined elements—a phenomenon known as synergy, as shown in Figure 1.

We will illustrate this therapeutic synergy by exhibiting the example of a male patient (#1). At first, he found CBT to be “demanding”, especially because, as he put it: “I wasn’t really aware of my own thinking”. This lack of insight made him feel that the homework (working with the ABC model) was “distressing” because he was not able to complete them, which made him feel insecure. The patient, however, gradually gained trust when the therapist asked questions that showed he was genuinely interested in him, and strived to understand how he was thinking and feeling, for example when he asked follow-up questions to clarify what the patient actually meant through Socratic questioning. This increased trust gave the patient “the safety” to “open up” about his negative feelings. He explains how the therapist then helped him become more aware of these feelings: “He often noticed that I was feeling something. Then he could say “I can see that you are feeling something”. And that made me more aware of that feeling. When he sees it, I see it too. He acknowledged the feeling I had, and that made me feel even stronger”.

The patient found it difficult at first to differentiate between thoughts and feelings, and the therapist showed him the cognitive diamond and ABC model. The patient then found it helpful to focus on bodily sensations: “It helped me when I became more aware of the physical sensations in my body. When I got a feeling in my body, I also got more aware of what was going on in my head. And then I started to pick up and understand my thoughts. (…) Then I manage to calm down, or realize that I’m not actually angry, but sad”. The ability to express emotions in therapy also helped him normalize and accept his own emotions: “If you feel them multiple times it gets … it’s not so scary anymore”. This increased insight and ability improved his relationship to his girlfriend: “First I dared to show emotions to my therapist. After a while I dared to show it at home too”.

After gaining this new insight, the patient expressed that he “suddenly saw the value of what we had been through”, referring to the homework he had previously experienced as demanding. Ultimately, he said that the homework was “very important” for his process of improving.

When the patient is asked what helped him the most—the therapeutic relationship, the specific therapeutic interventions or the new insights into his own thoughts and feelings—he answers: “I think it was the combination… I think most of it has to be there, and … It’s the whole package, I think”.

## 4. Discussion

This study was designed to gain insight into the process of improvement in CBT for MDD, and the main question was: How do patients experience their process of improvement? None of the patients explained their improvement as a linear experience, but rather as a complex and multilayered process. This is consistent with Levitt et al. who found that most qualitative studies indicate that patients experience improvement in therapy as a holistic lived experience, not as defined by singular forms or sequences of pattern identification [4]. Many patients considered the same elements as helpful, but often in different ways and at different stages of therapy. There also were substantial variations among the patients’ views on specific aspects of therapy, and elements that some patients found helpful others found either useless or even detrimental to their improvement. We suggest that this is not merely a consequence of individual preferences, but that the variation can be partly explained by the phenomenon of therapeutic synergy.

By using thematic analysis, we found three main elements to be significant in the typical process of improvement: the relationship between patient and therapist, the therapeutic interventions and the insight of the patient. These will now be discussed in more detail.

### 4.1. Elements in the Process of Improvement

#### 4.1.1. The Relationship

The relationship seems to be an important part of the healing process, but there is certainly not a general blueprint for success. The relationship needs to be “good enough”, but most patients expect nothing more than an interested, professional therapist that they feel they can trust. We also get the impression that successful therapeutic interventions and increased insight makes the relationship stronger because the patients experience that the therapist is helping them improve. A meta-review by Constantino et al. found that when patients in CBT were more optimistic about how much therapy would help them (high outcome expectation) this had a moderate effect on higher alliance quality, which had small-to-moderate effect on better post-treatment outcomes [27]. Successful interventions and increased insight in the early course of treatment could lead to increased trust in the therapists’ ability to help, which can create higher outcome expectations, which again could influence post-treatment outcome.

A couple of patients were at first a bit skeptical about their therapist because of a big difference in age. They both said that the therapist reminded them of older members of their family and expected their therapist to react and treat them in the same way. When they did not, this came as a surprise, and for one patient it became a moment of insight—perhaps an opportunity for insight through a corrective emotional experience. Again, this shows the complexity of the therapeutic relationship and is a reminder that transference is a phenomenon also relevant in CBT. It also shows that a discussion between patient and therapist about their similarities and differences is just as crucial in CBT as Levitt et al. found it to be in several approaches to therapy [4].

A few patients also said that they experienced better private relationships as a result of therapy, which was also found by De Smet et al. [10]. The mechanism for our patients seemed to be that the therapeutic relationship acted as a training ground for expressing emotion and that these emotions were accepted and normalized by the therapist. This made it easier for patients to accept their own emotions, their emotional confidence grew and this was transferred to their real-life relationships. This is an illustrative example of how CBT can utilize the interplay between the therapy room and the outside world to make the patients experience improvement.

Getting a glimpse of the therapist as a real person, not just as playing a professional role, is often mentioned by patients as strengthening the therapeutic alliance [4,28]. A qualitative review on self-disclosure by Henretty and Levitt shows the complexity of self-disclosure and underscores that it is important to identify early on the patients who feel burdened or uncomfortable by therapist disclosure [28]. In our study, we present an interesting example of the same therapist being described by two patients as being “personal” in her style and using self-disclosure. However, even though their description of the therapist is similar, their judgment is not. One patient says that this personal self-disclosure made her like and trust the therapist more, while the other lost trust in the therapist because it felt unprofessional to her. This shows the importance of an open dialogue with patients about their expectations and preferences and an adaptive therapist.

#### 4.1.2. The Interventions

Many therapeutic interventions were mentioned by patients to be important in the process of improvement. It seems to be just as much an issue of timing the intervention at the right moment in therapy as selecting the right intervention for the right patient. Additionally, just as with the relationship, we get the impression that patients seldom experience a causal connection between a therapeutic intervention and improvement.

The Socratic questioning seemed to both further the insight and strengthen the relationship. Patients found the therapists who used Socratic questioning to be genuinely interested, a therapist trait that most patients mentioned as essential. At the same time, they experienced increased insight because the questions stimulated alternative interpretations and thinking patterns, showing that one intervention can be experienced by a patient to have an effect on several elements of therapy. De Smet et al. showed that patients who got CBT especially valued that the therapy was interactive and that therapy offered insights through talking, specific questions and reflections, which is in line with our patients’ experiences, but De Smet et al. do not specify what types of questions were most stimulating [11]. Our results indicate that Socratic questioning stimulates our patients the most.

Some patients mentioned that it helped them improve when their therapist pointed out that they were thinking in a nonconstructive way. Often this was a gradual process, and sometimes it required other interventions like medication or behavior advice on sleep hygiene before the patients managed to internalize it. This shows that timing really matters when it comes to the success of an intervention, and supports the finding by Coutinho et al. that the timing of an intervention often decides if it strengthens or breaks the alliance [29].

Patients were also conflicted on the role of specific behavioral advice, most patients wanting more and a few patients wanting less advice. A few patients were provoked by getting advice and said that all they wanted was for the therapist to listen and accept their emotions. We speculated that this could have been avoided if the therapists had discussed the role of behavioral advice in therapy within the first few sessions. It is interesting to note that many patients said they got advice from their therapist during therapy, even though giving patients advice is not part of the manualized therapy. This may be because the therapists actually gave concrete advice, but we speculate that it may also be partly explained by a tendency patients have to take away from therapy what they need. If patients think they need advice to improve, as we have seen that many do, they may interpret the psychoeducation and homework as therapeutic advice. A qualitative study by Løvgren et al. found that depressed patients in PDT also felt they improved when they got suggestions from their therapist on how to do concrete practical things [30]. This indicates that even in PDT, where giving concrete advice is often viewed as counter-productive to therapy, patients may interpret the therapists’ suggestions as concrete advice and experience that the advice they are getting are helpful.

Homework assignments were generally viewed as important, but a few felt it put pressure on them that they felt they could not handle. We found this aspect of therapy to be difficult to many patients, especially at the beginning of therapy. That shows the importance of customized homework that feels relevant and achievable to the patient. This finding supports previous research, like the study by Barnes et al. were they found that homework was the main aspect of CBT that was described in a negative way [31]. Some felt distressed because it forced them to work through feelings and events between therapy sessions, and the authors theorize that the homework often conflicted with the patients’ coping mechanism of avoidance.

Giving patients autonomy and thereby responsibility for their own healing process is an important part of CBT, and seemed to be valued by most patients. This is in line with findings from other qualitative studies on CBT [9,11]. Autonomy was achieved by making the patients set the agenda in the sessions and bring therapeutic material to explore by doing their homework. Still, some patients felt they got too much responsibility when they were too depressed and confused to handle it, an important reminder for therapists to always consider whether the patients are able to handle the responsibility and make use of the autonomy they are given.

Some patients also emphasized the importance of medication and felt that antidepressants and non-addictive sleeping pills made them more able to participate constructively in therapy. Some of these patients even felt that starting antidepressants was a prerequisite for engaging in the cognitive aspects of therapy. The mechanism behind this is unknown to us, and further research on this particular topic is needed. Anyhow, it reminds us to always consider the potential benefits of combining psychotherapy and medication, a combination that has been shown to be superior to pharmacotherapy alone [6]. This should especially be considered when the patient is struggling to engage in the cognitive aspects of therapy.

Since every session has been videotaped and reviewed by a group of CBT supervisors, we know that all therapists have been following the basic treatment protocol. That makes it possible to reflect upon which therapeutic interventions the patients did not mention when asked about their improvement in therapy. None of the patients mention formulation of therapy goals, problem lists or case formulations as important for their improvement. They were not asked specifically about these interventions in the interviews, and it was never brought up by the patients. This is interesting in light of the study by Chen et al. where they found that the more patients improved their problem-solving skills in CBT, the more their depression decreased [32]. This was not mentioned specifically by our patients in the semistructured interviews. This discrepancy can possibly be explained by the discovery by De Smet et al. that a change in BDI does not always correlate with the patients’ experience of change [10]; it may contribute to a reduction in BDI, but not necessarily be experienced as an important part of the improvement. More research is needed to clarify this issue.

There seems to be big differences in which techniques and interventions patients remember from therapy, and especially in which techniques they still use in their daily life after therapy has ended. It seems that some patients use many and some patients use few techniques. This is in line with a qualitative study by a study by French et al. where they labeled patients as either “learners” or “talkers”, and noticed that the “learners” used more techniques from therapy than the “talkers” [33] post-therapy. Extra-therapeutic factors were seldom mentioned or emphasized in our material, and therefore not included in our study. That these are not mentioned by the patients does not mean they are not de facto important for the healing process.

#### 4.1.3. Insight

Increasing insight was the third element we identified as important in the patients’ process of healing. That gaining insight is an important step in improvement is also found by other studies, like that of De Smet et al. [10]. None of the patients clearly reported experiencing a phenomenon similar to what Tang and DeRubeis described as “sudden gains”, where they suggested a three-stage model with cognitive preparation, sudden gain and upward spiral [34], which has also been found in other schools of therapy, and in other psychiatric illnesses [35,36]. In contrast, nearly all our patients described that gaining insight was a gradual process.

Our patients reached new insights by numerous pathways, as is also found by Timulak et al. [37]. It seems that our patients gain insight in a much more diverse way than what was found by Elliot et al., where most of the insight in the CBT was gained by external reattribution of negative life events and greater self-assertion [38]. Some insights were mediated by discoveries within the therapeutic relationship, others by therapeutic interventions. Most often it was a combination of a good relationship and a well-suited intervention that the patient had enough insight to understand and improve from. For example, some patients struggled to organize their thoughts before they started medication and got frustrated with homework because they lacked in their ability to organize their thoughts and feelings. To organize them, they needed help from the therapist. However, the therapist could not help them before they were able to open up and show their feelings, which required a certain amount of safety and trust. This much needed trust, as we have seen, could also be gained via numerous pathways.

Our patients experienced few of the prototypic moments of insight as defined by Elliot et al., which consists of four elements: seeing something about oneself with a figurative eye; seeing connections, reasons, causes, categorizations or parallels; a feeling of suddenness, mental clicking or surprise and the feeling of discovering something not previously known [38]. Very few patients remembered exact scenes or interventions from their therapy that were major discoveries or breakthroughs, probably because the process as a whole is more important than the specific details—at least in the patient’s experience.

This can hopefully inspire therapists to think of gaining insight as a process where consistency, patience and stability are the most important ingredients, and not expect immediate results from even their wisest comments.

#### 4.1.4. Therapeutic Synergy

There still does not seem to be any magical bullet in therapy. Like one of the patients put it: “It’s the whole package”. In this article we tried to open this package to examine its main components. We also suggest that the process of improvement in CBT is best understood as a synergetic process. Very few patients could point to any particular event or intervention in therapy that gave them a definitive breakthrough or major insight. It seems like gaining insight, just like the development of a therapeutic relationship and utilization of therapeutic interventions, is experienced as a slow and multifaceted process where each element strengthens the others.

As explained by Figure 1, we found that an intervention often not only increased insight, but also strengthened the therapeutic relationship. We hypothesized that the relationship is strengthened because the patient feels helped by the therapist. This theory is supported by the discovery by Tang and DeRubeis that the therapeutic relationship seemed to get better after patients experienced sudden gains [34]. The strengthened relationship could in turn make it easier for the patient to open up and access difficult emotions in therapy, as already mentioned. A stronger relationship and increasing trust will open up new possibilities for reaching new insight. With increased insight, the patient may be open for new kinds of interventions that were ineffective at an earlier stage in therapy. For example, an intervention like behavioral advice will often fail to help the patient if trust is not established, and homework is just distressing to a patient who lacks the interventional skills or basic insight to complete it. This is an ongoing process throughout therapy, and obviously not a straight-forward process; both the relationship and insight can have setbacks.

### 4.2. Do Our Patients’ Reports Match Becks Theories?

When we compare how patients explain and experience their improvement process with the theoretical rationale that is the foundation for CBT, we find great overlap [19]. Most patients who were feeling better said they owed this to a change in their cognition, attitude or their assumptions about themselves. Although none of the patients used the word “restructuring”, it was evident that they often felt they improved when they changed their thoughts about themselves, the world and/or the future. We also showed that this cognitive restructuring, like Aaron Beck wrote, could happen through several different techniques. The core improvement techniques presented by Beck are the same mentioned by the patients. Monitoring of negative, automatic thoughts; examining of evidence for and against these thoughts; substituting distorted thoughts with more reality-oriented interpretations; recognizing the connections between cognition, affect and behavior; and identifying and altering dysfunctional beliefs still seem to be what the patients experience as the main mechanism of positive change in CBT.

Although none of the patients used the expression “metacognition”, they appeared to experience an increased ability to think about their own thoughts, which facilitated improvement. Some of the patients in this study mention increased emotional awareness as central to their improvement, both related to increased self-acceptance and improved interpersonal relationships. Beck’s theories emphasize cognitive rather than emotional change as primary to improvement. However, both emotional and physical experiences are central in interventions like the cognitive diamond and the ABC model, and Beck himself underlined the importance of changing the emotional response patterns, albeit through a change of cognitions. In sum, our study strengthens the theoretical rationale behind CBT.

### 4.3. Strengths and Limitations

From a methodological standpoint, it is impossible to differentiate between correlation and causality in a group consisting of just ten patients. Thus, we decided to focus on the process of improvement as retrospectively experienced by the patients. Obviously, our study consists of a relatively small sample size, so these results did not apply directly to other individual patients. The mean age of patients was 36 years, and readers should not automatically suppose that adolescents or elderly patients would experience therapy in a similar way. However, we still think our results have some implications of clinical value. Readers are invited to transfer our results to their own context and clinical situation—with a cautious reminder about the dangers of generalizations.

The results obtained in our material do not contradict the findings from big meta-analyses of the mechanisms of several different approaches to therapy [3,4,5]. Our patients seem to experience therapy in a similar way as described in these studies, albeit not identical. Due to the design of our study, and the synergistically entangled relationship between common and specific factors, we cannot say exactly which elements are specific for CBT and which are not.

Most qualitative studies conceptualize their findings in different ways, resulting in a myriad of categories and theories. The vast difference between patients’ experiences of therapy raises the question as to whether it is even possible to fully represent this diversity and at the same time unify these unique experiences into broader themes and categories. We used a strict method of thematic analysis to do this, and the material was reviewed by several authors with different education and background.

We tried not to use theoretical terms or concepts from the cognitive tradition to interpret the patients’ experiences, and tried to be as open minded as possible. We are aware that a lot of the phenomena we discuss may be interpreted differently than we have done, and that we may have tended, unintentionally, to interpret what the patients said from the perspective of the treatment they got (CBT). Two of the authors are CBT clinicians, which may explain such a bias, but it is a strength that we also included a PDT clinician and that the three other authors did not have a specific orientation.

## 5. Conclusions

We identified three elements that were relevant to the process of improvement for all patients: the therapeutic relationship, the therapeutic interventions and gaining insight. There is a dynamic interrelationship and synergy between these elements that may explain why patients often considered the same elements as helpful, but in different ways and at different stages of therapy. By understanding the synergies and interrelationship between the elements, we hope that therapists can utilize them even better. We also found substantial variations among the patients’ views on specific aspects of therapy; what some patients found helpful, others found useless or even detrimental to their improvement. Our study is a reminder of the importance of always being attentive to the individual patient’s process of improvement.

## Figures and Tables

**Figure 1 ijerph-18-02292-f001:**
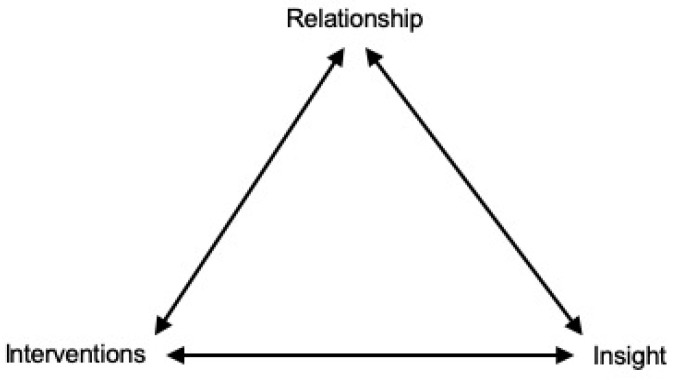
Dynamic interrelationship and synergy between elements of therapy. For example, a single intervention was not only experienced to increase insight, but also to strengthen the relationship. The strengthened relationship could in turn make it easier for the patient to open up and access difficult emotions in therapy. This will further make it possible for the patient to reach new insights, and also make the patient open to new therapeutic interventions. This is an ongoing process throughout therapy.

**Table 1 ijerph-18-02292-t001:** This table shows the questions used in the semistructured interview of patients.

Main Theme	Specific Questions (Some Examples)
How did you experience being in therapy?	Can you tell about the treatment you got? How was your relationship to the therapist? How did the treatment suit your needs?
What contributed to improvement in your therapy?	In what way did it contribute? How did you experience the therapy? What changed during the therapy? In what way did the changes help you? How did you notice improvement? Can you describe a moment where you really experienced that the therapy was helpful?
Was there anything in therapy that you experienced as not being helpful?	In what way was it not helpful? Why? What should have been different? If you get problems later, will you seek help again? Why/why not?
How did therapy influence relationships and other important aspects of your life?	How did it affect your relationships in family/friends/work/school?
Was there anything from therapy that you can use today or in the future?	When do you use it? In what way?

**Table 2 ijerph-18-02292-t002:** Description of the treatment manual.

**Sessions 1–3**	Building therapeutic alliance, socialization to the model, basic case formulation, problem list and goals
**Sessions 4–16**	Explore link between thoughts, feelings, physical sensations and actions through the ABC/Cognitive Diamond model and Socratic questioning. Identifying thinking traps. Expand case formulation. Explore alternative thinking and behavioral patterns.
**Sessions 17–19**	Consolidation and relapse prevention

**Table 3 ijerph-18-02292-t003:** Elements and important aspects, as experienced by patients, are summarized.

Elements	Important Aspects
Relationship	Chemistry
	Trust
	Interest
	Transference
	Becoming personal
	Being liked
Interventions	Socratic questioning
	Cognitive diamond and the ABC model
	Discussing thinking traps
	Advice on behavior
	Homework assignments
	Autonomy/responsibility
	Medication
Insight	Raising awareness of feelings, thoughts, bodily sensations and behaviors.
	Gaining ability to create distance from and to question thoughts.
	Focusing more on the positive aspects of the self.

## Data Availability

Data sharing not applicable.

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
