# Peer review of "The Synergistic Process of Improvement in Cognitive Behavioral Therapy for Major Depression"

_ijerph, 2021, doi:10.3390/ijerph18052292_

Round 1
Reviewer 1 Report
Dear Authors.
An interesting work in which you highlight the effectiveness of cognitive-behavioral therapies in major depression.
Now I will comment on some aspects to improve in some sections of the article.
Introduction: In the introduction you mention some studies but I would add more systematic reviews and meta-analyses on these topics since the theoretical framework is more robust than with some randomly chosen studies.
In the section Design, Ethics and Data Collection, it is commented that patients were randomly assigned to one therapy or another. Please explain if this aspect is ethical taking into account that randomness cannot be a criterion for choosing therapy for a patient.
As for the participants, it should be included in the limitations that it is a rather small sample and that it is a study carried out with middle-aged patients. In other words, the results could not be generalized to other age groups.
I also recommend that they use the DSM-5 to diagnose personality disorder. Regarding personality disorders in DSM-5 I recommend citing the following article:
Ozamiz-Etxebarria N, Ortiz-Jauregi MA, Escobar JI. Validating the Alternative DSM-5 Criteria for Personality Disorders: A Study in the Basque Region of Spain. J Nerv Ment Dis. 2019;207(3):199-202. doi:10.1097/NMD.0000000000000948
In the therapists and treatment section please specify if the therapists had a psychology degree.
One of the questions I asked myself when reading the paper was whether patients had to pay for the therapies performed. This aspect can influence the perception about the relationship with the therapist and it would be important to underline if this aspect has been taken into account or if they think it can influence the effectiveness of the therapy and the patient's perception towards the therapist.
Congratulations for the work.
Author Response
Reviewer reports:
Reviewer #1
Dear Authors.
An interesting work in which you highlight the effectiveness of cognitive-behavioral therapies in major depression.
Now I will comment on some aspects to improve in some sections of the article.
1: Introduction: In the introduction you mention some studies but I would add more systematic reviews and meta-analyses on these topics since the theoretical framework is more robust than with some randomly chosen studies.
Answer: We have included three meta-analyses in the introduction (McPherson et al, Levitt et a, Timulak et al). Please read our answer on this topic in our comments to reviewer #3.
2) In the section Design, Ethics and Data Collection, it is commented that patients were randomly assigned to one therapy or another. Please explain if this aspect is ethical taking into account that randomness cannot be a criterion for choosing therapy for a patient.
Answer: We agree that this is an important aspect to consider in randomized controlled trials. However, in the present study we compare CBT and PDT which have been found to be equally effective in treating major depression. Furthermore, the patients are informed that they will be randomized to CBT or PDT before they sign the informed consent. Patients with strong preferences for either CBT or PDT who do not want to be randomly assigned to one of the treatment approaches will not be included in the study.
As for the participants, it should be included in the limitations that it is a rather small sample and that it is a study carried out with middle-aged patients. In other words, the results could not be generalized to other age groups.
Answer: We agree, and have included these comments in our paper (Line 720-730).
3) I also recommend that they use the DSM-5 to diagnose personality disorder. Regarding personality disorders in DSM-5 I recommend citing the following article:
Ozamiz-Etxebarria N, Ortiz-Jauregi MA, Escobar JI. Validating the Alternative DSM-5 Criteria for Personality Disorders: A Study in the Basque Region of Spain. J Nerv Ment Dis. 2019;207(3):199-202. doi:10.1097/NMD.0000000000000948
Answer: We agree with the reviewer that it would be interesting to use the DSM-5 to diagnose personality disorder. However, when we started to include patients, the manual for diagnosing DSM-5 personality disorders was not translated to Norwegian.
4) In the therapists and treatment section please specify if the therapists had a psychology degree.
Answer: In accordance with the reviewer’s suggestion we have specified whether the therapists had a psychology degree (Line 182).
5) One of the questions I asked myself when reading the paper was whether patients had to pay for the therapies performed. This aspect can influence the perception about the relationship with the therapist and it would be important to underline if this aspect has been taken into account or if they think it can influence the effectiveness of the therapy and the patient's perception towards the therapist.
Answer: We agree and have described this in the Material and methods section (Therapist and treatment) (Line 179)
Reviewer 2 Report
Abstract → Concise and covers the main aspects. A small detail is that Background appears in bold (line 12), unlike Method, Results and Conclusion.
Introduction → There is a good review of studies related to the article, including publications of the last 14 years, please include more reference from the last 5 years
The introduction is clearly worded. I found only 2 small wording details, in line 31 it appears "is highly needed" and it should be "are highly needed", since it refers to effective methods to prevent episodes of major depression, therefore with the plural "are". Also in line 38 the word "in" is missing in the phrase "has grown the last decades", which should say "has grown in the last decades".
Materials and method → The sample is described quite well, since the general characteristics are given and the presence of other diagnoses and the dropout rate are also detailed. The description of the therapists, as well as the process of applying the therapy are fully described. The interview application and analysis procedure is detailed, and considers the various therapeutic orientations of the authors. Here from lines 224 to 237 I got a little lost ... are the requirements of the journal? Please check
Results → The results are concrete and presented in an orderly manner.
Please include identification code in the patient's reference.
Discussion → The discussion is clear, orderly and related studies are cited. In line 562 there is a letter “s”, where it says “an interventions”, as it should say “an intervention”
References → In the references, some journals appear with the full name and others with the abbreviated name. In the rules of the journal it says that the names must appear abbreviated.
Author Response
Reviewer #2
1)
Abstract → Concise and covers the main aspects. A small detail is that Background appears in bold (line 12), unlike Method, Results and Conclusion.
Answer: Corrected, and made uniform.
2)
Introduction → There is a good review of studies related to the article, including publications of the last 14 years, please include more reference from the last 5 years
The introduction is clearly worded. I found only 2 small wording details, in line 31 it appears "is highly needed" and it should be "are highly needed", since it refers to effective methods to prevent episodes of major depression, therefore with the plural "are". Also in line 38 the word "in" is missing in the phrase "has grown the last decades", which should say "has grown in the last decades".
Answer:
Unfortunately, we have not found further qualitative studies on CBT from the last five years that appear relevant to this paper. We have searched in several databases. We have however added a paper from 2016 (French et. al) in the discussion that featured in the meta-analysis referred to in the paper (McPherson et al). We have also added a paper from 2019 by (Cuijpers et al) in the introduction.
Wording details/grammar corrected (line 32 and 40).
3) Materials and method → The sample is described quite well, since the general characteristics are given and the presence of other diagnoses and the dropout rate are also detailed. The description of the therapists, as well as the process of applying the therapy are fully described. The interview application and analysis procedure is detailed, and considers the various therapeutic orientations of the authors. Here from lines 224 to 237 I got a little lost ... are the requirements of the journal? Please check
Answer: This was due to some of the template-text being included in the manuscript. We are very sorry that this happened.
4)
Results → The results are concrete and presented in an orderly manner.
Please include identification code in the patient's reference.
Answer: We appreciate this suggestion, and patient identification has been added. Each patient is numbered #1-#10 in the text in the results-section.
5)
Discussion → The discussion is clear, orderly and related studies are cited. In line 562 there is a letter “s”, where it says “an interventions”, as it should say “an intervention”
Answer: Corrected.
6)
References → In the references, some journals appear with the full name and others with the abbreviated name. In the rules of the journal it says that the names must appear abbreviated.
Answer: We have corrected this as well.
Reviewer 3 Report
This is a qualitative study principally aimed at investigating how patients suffering from MDD experience improvement after CBT therapy.
The qualitative study design is quite sound, where sessions were videotaped and reviewed. The coders have different psychological orientations which is another strength. In this sense, the study seems to be quite well executed at a methodological level.
That said, when reading the paper it is very unclear to me what this study adds to the current literature. The introduction is not well formulated. The authors describe a lot of qualitative research from psychotherapy, including CBT, but then conclude that the literature on patients’ experiences of CBT for MDD is scarce and that is why they conduct their study. This seems counterintuitive to me because from reading the introduction I had the sense that there is quite a lot of qualitative literature. In this sense, the authors need to make the case for their study. What is it addressing specifically? What are the research gaps that are trying to cover? What do we not know that merits this study to be conducted? This is really important because it will help as well to structure the discussion and distinguish what is important and what is not (I’ll talk about that later as well).
Some key information on the method is missing:
- More detail is needed about the interview. Line 135-149 would benefit to be separated into a specific section on the data collection and the interview process. Add questions or prompts into a table or figure to clarify what the interview entailed.
- More description into the treatment is needed: add table with description of the protocol of sessions. Also, how many sessions was the protocol? Or what was the average of sessions offered?
Results section is very long. I understand this is needed given the detailed collection of information from the interviews. However, a table summarizing the main findings would be important to facilitate reading and would help to reduce the length on the results sections.
In the point 3.2, I think you are referring to therapeutic techniques (or CBT techniques) instead of interventions, which is very confusing because typically intervention is synonym of treatment. Please change the term across the manuscript.
I understand that you have built this manuscript from a template from the journal but you need to read carefully and remove all standard text from the template. For example in lines 424-426 and 731-734 you have text from the template. Please ensure this is removed and that there are not more of these across the paper.
Discussion: The discussion is extremely lengthy, and I cannot distinguish what is more important from what it’s not that important, which is due to how the study is framed. Because is unclear what this study is adding to the current literature, I cannot assess if what you are discussing is really relevant or is just a repetition from the results. You need to do an exercise of being more concise on the discussion and to the point. If something is not conclusive or unclear is better to not add much detail about it.
Lines 610-615: These are assumptions based on the absence of information but it doesn't mean it was not experienced. It could depend on how these techniques were framed in therapy and patients not being able to articulate this technique, rather than just concluding it contributed less to the experience of improvement.
MINOR COMMENTS
Methods:
Line 125: Describe a bit more about the type of services offered in the clinics. Are they public or private? is it secondary care services? Type of patients treated at these services?
Participants, Line 154-162:
- Include inclusion criteria for the main study
- Include interpretation scores for Hamilton Scale, what’s 15,4 indicating in terms of severity levels?
- Line 160-161: were all other participants (apart from the one with personality disorder) diagnosed only with MDD and not other disorders?
Discussion, Line 612-614: Was this found through a questionnaire or through qualitative interviews? because in the former they would have been explicitly asked, which is very different from expecting patients to mention it. Please explain.
Author Response
Review 3
This is a qualitative study principally aimed at investigating how patients suffering from MDD experience improvement after CBT therapy.
The qualitative study design is quite sound, where sessions were videotaped and reviewed. The coders have different psychological orientations which is another strength. In this sense, the study seems to be quite well executed at a methodological level.
1)
That said, when reading the paper it is very unclear to me what this study adds to the current literature. The introduction is not well formulated. The authors describe a lot of qualitative research from psychotherapy, including CBT, but then conclude that the literature on patients’ experiences of CBT for MDD is scarce and that is why they conduct their study. This seems counterintuitive to me because from reading the introduction I had the sense that there is quite a lot of qualitative literature. In this sense, the authors need to make the case for their study. What is it addressing specifically? What are the research gaps that are trying to cover? What do we not know that merits this study to be conducted? This is really important because it will help as well to structure the discussion and distinguish what is important and what is not (I’ll talk about that later as well).
Answer: We agree that our novel contribution should be more clearly stated, and have revised the introduction to clarify what this study contributes with. The synergistic process of improvement as experienced by patients has, to our knowledge, not been described in other qualitative studies in CBT. We realize that this point is understated in our article. We therefore added the word “synergistic”to the title and further made synergistic processes the definite focus of the discussion.
However, we do find that the there is a substantial lack of qualitative research concerning face-to-face individual CBT for depressed adult patients. As mentioned in the article, all of the qualitative meta-syntheses/meta-analyses have had to merge change mechanisms from different therapeutic orientations together – because the literature is scarce. The same applies – for example – for the study done by De smet and Nilsson. In our study, we do not merge PDT and CBT, but take an in-depth look at the mechanism in CBT and how they are experienced by patients. We have tried to get this point across in a better way in our revised edition of the introduction (lines 116 – 126).
2)
Some key information on the method is missing:
- More detail is needed about the interview. Line 135-149 would benefit to be separated into a specific section on the data collection and the interview process. Add questions or prompts into a table or figure to clarify what the interview entailed.
Answer: We have separated this into a specific section (2.2) and added questions into a table. See line 158 and Table 1.
3)
- More description into the treatment is needed: add table with description of the protocol of sessions. Also, how many sessions was the protocol? Or what was the average of sessions offered?
Answer: We have added more information about the therapy protocol in the text (line 187 – 194). As stated, all patients were offered 16 + 3 sessions. We have also added a reference to our recently published study protocol (line 135).
4)
Results section is very long. I understand this is needed given the detailed collection of information from the interviews. However, a table summarizing the main findings would be important to facilitate reading and would help to reduce the length on the results sections
Answer: We understand that this article is a long read, and appreciate the suggestion to add a table to facilitate reading. A table has been added, see line 268.
5)
In the point 3.2, I think you are referring to therapeutic techniques (or CBT techniques) instead of interventions, which is very confusing because typically intervention is synonym of treatment. Please change the term.
Answer: We appreciate this comment and have specified what we mean by the term “intervention” in the text in the 3.2 section (line 325). We believe that intervention is a term that can include techniques, assignments and medication – but realize that we have not clarified this in the text. We hope you find this solution satisfying.
6)
I understand that you have built this manuscript from a template from the journal but you need to read carefully and remove all standard text from the template. For example in lines 424-426 and 731-734 you have text from the template. Please ensure this is removed and that there are not more of these across the paper.
Answer: We are sorry for this unfortunate mistake. It has been corrected.
7)
Discussion: The discussion is extremely lengthy, and I cannot distinguish what is more important from what it’s not that important, which is due to how the study is framed. Because is unclear what this study is adding to the current literature, I cannot assess if what you are discussing is really relevant or is just a repetition from the results. You need to do an exercise of being more concise on the discussion and to the point. If something is not conclusive or unclear is better to not add much detail about it.
Answer: We clearly agree with the reviewer and have thoroughly revised the discussion section. Most importantly we have revised the discussion to make it more concise and to the point. We have further focused more pointedly on the concept of synergy. We have tried to highlight in the text when our study supports or contradicts other studies.
8)
Lines 610-615: These are assumptions based on the absence of information but it doesn't mean it was not experienced. It could depend on how these techniques were framed in therapy and patients not being able to articulate this technique, rather than just concluding it contributed less to the experience of improvement.
Answer: Agreed. We have changed this accordingly. (line 619 – 634)
MINOR COMMENTS
Methods:
Line 125: Describe a bit more about the type of services offered in the clinics. Are they public or private? is it secondary care services? Type of patients treated at these services?
Participants, Line 154-162:
- Include inclusion criteria for the main study
- Include interpretation scores for Hamilton Scale, what’s 15,4 indicating in terms of severity levels?
- Line 160-161: were all other participants (apart from the one with personality disorder) diagnosed only with MDD and not other disorders?
Discussion, Line 612-614: Was this found through a questionnaire or through qualitative interviews? because in the former they would have been explicitly asked, which is very different from expecting patients to mention it. Please explain.
Answer: We have added some more information about the outpatient clinics (line 131-135), included inclusion and exclusion criteria (line 162-167), Hamilton scale interpretation (172). Regarding diagnosis, some had anxiety disorders, all of which were considered to be mild, which is now specified in the text (174).
Round 2
Reviewer 3 Report
The authors have addressed most of my comments and I feel the paper is improved. I’d like to congratulate the authors for the new additions and the changes.
There are still few minor issues that could be solved before publication. I have only included below those comments where my concerns were not fully addressed, and removed the ones I’m happy with.
RESPONSES
3)
- More description into the treatment is needed: add table with description of the protocol of sessions. Also, how many sessions was the protocol? Or what was the average of sessions offered?
Answer: We have added more information about the therapy protocol in the text (line 187 – 194). As stated, all patients were offered 16 + 3 sessions. We have also added a reference to our recently published study protocol (line 135).
Even though the authors report having described the number of sessions, that is not included in the paper. Please correct this issue.
Was the therapy manualized? If so, please describe the order of the therapeutic components. This will help to understand the results of the interviews later on, since it is not obvious to readers (a table might help). If was not manualized, please state so and describe, at a general level, how the treatment was meant to be individualized for each patient.
- Line 160-161: were all other participants (apart from the one with personality disorder) diagnosed only with MDD and not other disorders?
Answer: We have added some more information about the outpatient clinics (line 131-135), included inclusion and exclusion criteria (line 162-167), Hamilton scale interpretation (172). Regarding diagnosis, some had anxiety disorders, all of which were considered to be mild, which is now specified in the text (174).
Please be more precise, how many had diagnosis of anxiety disorders and what were the specific diagnoses? This is important to understand the characteristics of the 10 patients
NEW COMMENTS
Line 503-512. I don't get this point. Your finding relates to the fact that good relationship is also a product of a successful treatment and increased insight. I don't think that speaks to the outcome expectations but to the perceived benefit and they're two different things. Hence, I don't think the findings from Constantino's meta-review are talking about the same thing. Please clarify the discussion of this finding or re-elaborate.
Line 512-513. This sentence is a bit confusing. What do you mean by “making therapy effective from the start”? Either re-elaborate or remove.
Author Response
Reviewer reports:
Reviewer # 3
The authors have addressed most of my comments and I feel the paper is improved. I’d like to congratulate the authors for the new additions and the changes.
There are still few minor issues that could be solved before publication. I have only included below those comments where my concerns were not fully addressed, and removed the ones I’m happy with.
RESPONSES
3)
Previous comment from reviewer: More description into the treatment is needed: add table with description of the protocol of sessions. Also, how many sessions was the protocol? Or what was the average of sessions offered?
Previous answer: We have added more information about the therapy protocol in the text (line 187 – 194). As stated, all patients were offered 16 + 3 sessions. We have also added a reference to our recently published study protocol (line 135).
New comment: Even though the authors report having described the number of sessions, that is not included in the paper. Please correct this issue.
Was the therapy manualized? If so, please describe the order of the therapeutic components. This will help to understand the results of the interviews later on, since it is not obvious to readers (a table might help). If was not manualized, please state so and describe, at a general level, how the treatment was meant to be individualized for each patient.
New answer: We have now added number of sessions to the paper (line 136-138). The therapy was manualized and we have now added a table to provide the protocol (189-198).
Previous comment from reviewer: Line 160-161: were all other participants (apart from the one with personality disorder) diagnosed only with MDD and not other disorders?
Prev, answer: We have added some more information about the outpatient clinics (line 131-135), included inclusion and exclusion criteria (line 162-167), Hamilton scale interpretation (172). Regarding diagnosis, some had anxiety disorders, all of which were considered to be mild, which is now specified in the text (174).
New comment from reviewer: Please be more precise, how many had diagnosis of anxiety disorders and what were the specific diagnoses? This is important to understand the characteristics of the 10 patients
New answer: We have now been more specific on the diagnoses, line (168-173).
NEW COMMENTS
Line 503-512. I don't get this point. Your finding relates to the fact that good relationship is also a product of a successful treatment and increased insight. I don't think that speaks to the outcome expectations but to the perceived benefit and they're two different things. Hence, I don't think the findings from Constantino's meta-review are talking about the same thing. Please clarify the discussion of this finding or re-elaborate.
Answer: We have re-elaborated this to clarify the discussion (512-518).
Line 512-513. This sentence is a bit confusing. What do you mean by “making therapy effective from the start”? Either re-elaborate or remove.
Answer: This sentence has been removed.